# Task-evoked metabolic demands of the posteromedial default mode network are shaped by dorsal attention and frontoparietal control networks

Godber M Godbersen[1,2], Sebastian Klug[1,2], Wolfgang Wadsak[3,4], Verena Pichler[3,5], Julia Raitanen[3,6,7], Anna Rieckmann[8,9,10,11], Lars Stiernman[8,10], Luca Cocchi[12,13], Michael Breakspear[14,15], Marcus Hacker[3], Rupert Lanzenberger[1,2], Andreas Hahn[1,2]*

[1]Department of Psychiatry and Psychotherapy, Medical University of Vienna, Vienna, Austria; [2]Comprehensive Center for Clinical Neurosciences and Mental Health (C3NMH), Medical University of Vienna, Vienna, Austria; [3]Department of Biomedical Imaging and Image-guided Therapy, Division of Nuclear Medicine, Medical University of Vienna, Vienna, Austria; [4]Center for Biomarker Research in Medicine (CBmed), Graz, Austria; [5]Department of Pharmaceutical Sciences, Division of Pharmaceutical Chemistry, University of Vienna, Vienna, Austria; [6]Ludwig Boltzmann Institute Applied Diagnostics, Vienna, Austria; [7]Department of Inorganic Chemistry, Faculty of Chemistry, University of Vienna, Vienna, Austria; [8]Department of Integrative Medical Biology, Umeå University, Umeå, Sweden; [9]Department of Radiation Sciences, Umeå University, Umeå, Sweden; [10]Umeå Center for Functional Brain Imaging, Umeå University, Umeå, Sweden; [11]The Munich Center for the Economics of Aging, Max Planck Institute for Social Law and Social Policy, Munich, Germany; [12]Clinical Brain Networks Group, QIMR Berghofer Medical Research Institute, Brisbane, Australia; [13]School of Biomedical Sciences, Faculty of Medicine, University of Queensland, Brisbane, Australia; [14]School of Medicine and Public Health, College of Health, Medicine and Wellbeing, The University of Newcastle, Callaghan, Australia; [15]School of Psychological Sciences, College of Engineering, Science and Environment, The University of Newcastle, Callaghan, Australia

*For correspondence:
andreas.hahn@meduniwien.ac.at

**Abstract** External tasks evoke characteristic fMRI BOLD signal deactivations in the default mode network (DMN). However, for the corresponding metabolic glucose demands both decreases and increases have been reported. To resolve this discrepancy, functional PET/MRI data from 50 healthy subjects performing Tetris were combined with previously published data sets of working memory, visual and motor stimulation. We show that the glucose metabolism of the posteromedial DMN is dependent on the metabolic demands of the correspondingly engaged task-positive networks. Specifically, the dorsal attention and frontoparietal network shape the glucose metabolism of the posteromedial DMN in opposing directions. While tasks that mainly require an external focus of attention lead to a consistent downregulation of both metabolism and the BOLD signal in the posteromedial DMN, cognitive control during working memory requires a metabolically expensive BOLD suppression. This indicates that two types of BOLD deactivations with different oxygen-to-glucose index may occur in this region. We further speculate that consistent downregulation of the two signals is mediated by decreased glutamate signaling, while divergence may be subject to

active GABAergic inhibition. The results demonstrate that the DMN relates to cognitive processing in a flexible manner and does not always act as a cohesive task-negative network in isolation.

## Editor's evaluation

This important study advances our understanding of the metabolic and hemodynamic underpinnings of different brain networks. The evidence is convincing, drawn from multiple datasets and including simultaneous fMRI and PET. The study will be of interest to neuroscientists and researchers who use functional neuroimaging tools to study brain activity.

## Introduction

Large-scale brain networks progressively integrate sensory input to enable complex cognitive processes and behaviors in accordance with internal goals (*Cocchi et al., 2013*). The default mode network (DMN) was originally termed 'task-negative' due to decreased blood oxygen level dependent (BOLD) signal ('deactivation', see Methods for full description) during the performance of tasks requiring an external focus of attention as compared to rest (*Raichle, 2015*; *Raichle et al., 2001*). Processing of external stimuli in turn increases the BOLD signal in 'task-positive' networks including the dorsal attention (DAN) and the frontoparietal network (FPN). While the DAN is mainly engaged when attention to sensory information is required, such as in visuo-spatial reasoning (*Fox et al., 2005*), the FPN mediates cognitive control across various task conditions (*Cole et al., 2013*) such as the maintenance and manipulation of information also in the absence of external sensory stimuli (*Scolari et al., 2015*). Because of the low (or anti-) correlation in BOLD signals between task-positive and default mode networks at resting-state, it has long been assumed that an antagonism between the DMN and other large-scale networks represents a general characteristic of brain functioning (regardless if computed with or without global signal regression *Buckner and DiNicola, 2019*; *Murphy and Fox, 2017*). However, the DMN can also be activated during complex cognitive tasks (e.g., memory recollection, abstract self-generated thought; *Hearne et al., 2015*; *Smallwood et al., 2021*). This DMN engagement is thought to facilitate between-network integration, instead of being a segregated network alone (*Fornito et al., 2012*; *Leech et al., 2012*; *Smallwood et al., 2021*). Moreover, some brain regions such as the posterior cingulate cortex (PCC)/precuneus (part of the posteromedial DMN) have been suggested to play a key role in across-network integration (*Cocchi et al., 2013*). Thus, it has been suggested that task-positive attention/control networks and the DMN can flexibly switch between cooperative and antagonistic patterns to adapt to the task context at hand (*Cocchi et al., 2013*). Despite these advancements in the organization of brain networks, our understanding of the underlying metabolic demands of these context-specific neuronal processes is limited (*Goyal and Snyder, 2021*).

In this context, functional PET (fPET) imaging represents a promising approach to investigate the dynamics of brain metabolism. fPET refers to the assessment of stimulation-induced changes in physiological processes such as glucose metabolism (*Hahn et al., 2016*; *Villien et al., 2014*) and neurotransmitter synthesis (*Hahn et al., 2021*) in a single scan. The temporal resolution of this approach of 6–30 s (*Rischka et al., 2018*) is considerably higher than that of a conventional bolus administration. This is achieved through the constant infusion of the radioligand, thereby providing free radioligand throughout the scan that is available to bind according to the actual task demands. Here, the term 'functional' is used in analogy to fMRI, where paradigms are often presented in repeated blocks of stimulation, which can subsequently be assessed by the general linear model.

Studies using simultaneous fPET/fMRI have shown a strong spatial correspondence between the BOLD signal changes and glucose metabolism in several task-positive networks and across various tasks requiring different levels of cognitive engagement (*Hahn et al., 2020*; *Hahn et al., 2016*; *Jamadar et al., 2019*; *Rischka et al., 2018*; *Stiernman et al., 2021*; *Villien et al., 2014*). However, also regional differences in activation patterns have been observed previously between both modalities in these and previous studies (*Wehrl et al., 2013*). Moreover, a dissociation between BOLD changes (negative) and glucose metabolism (positive) has recently been observed even in the same region of the DMN during a working memory task (*Stiernman et al., 2021*), namely the posteromedial default mode network (pmDMN). In contrast, simple visual and motor tasks elicited a negative

metabolic response in this area (*Hahn et al., 2018*). These findings suggest that distinct underlying metabolic processes support state-specific BOLD signal changes in the DMN (*Goyal and Snyder, 2021*). However, the consistency and functional specialization of neuronal and metabolic interactions between default mode and task-positive networks is unclear. Specifically, it is unknown whether the observed dissociation between patterns of metabolism and BOLD changes in the DMN generalizes for complex cognitive tasks, and whether this in turn depends on the brain networks supporting the task performance and their interaction with the DMN.

To address these open questions, we employed functional PET/MR imaging with [¹⁸F]FDG during rest and performance of a visuo-spatial task requiring a broad range of cognitive functions primarily supported by the dorsal attention network (*Hahn et al., 2020*; *Klug et al., 2022*). The results were then compared to previously published data from tasks of varying complexity (working memory, eyes opening and finger tapping), primarily involving different brain networks (FPN, visual and motor, respectively). The task response of the posteromedial DMN was the main focus of interest, due to its key role in integrating specialized large-scale brain networks (*Cocchi et al., 2013*; *Leech et al., 2012*). The combination of different imaging modalities and cognitive tasks sheds further light on the interaction across brain networks involved in external attention and cognitive control as well as on the corresponding metabolic underpinnings of the BOLD signal.

## Results

Simultaneous fPET/fMRI data and arterial blood samples were acquired from 50 healthy participants during the performance of the video game Tetris, a challenging cognitive task requiring rapid visuo-spatial processing and motor coordination (*Hahn et al., 2020*; *Klug et al., 2022*). From this dataset

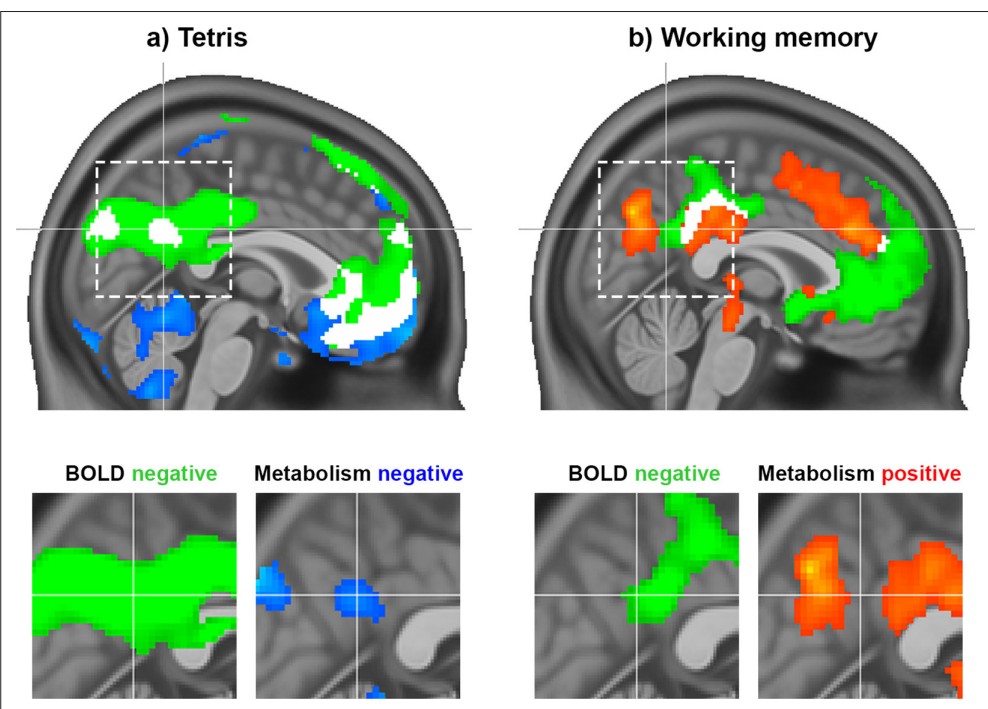

**Figure 1.** Task responses observed during the Tetris (DS1) and working memory tasks (DS2) (*Stiernman et al., 2021*). (**a**) The Tetris task employed in the current work elicited a negative response in the pmDMN for both the BOLD signal (green) and CMRGlu (blue). (**b**) For comparison, previously published results from a working memory manipulation task were also included, which showed a dissociation between BOLD and glucose metabolism in the PCC, that is, negative BOLD response (green) vs. increased metabolism (red). White clusters represent the intersection of significant CMRGlu and BOLD signal changes, irrespective of direction. Note, that also relevant differences between both imaging parameters can be observed, such as decreased CMRGlu in the cerebellum (in both datasets), without changes in the BOLD signal. The dashed rectangle indicates the zoomed section of the PCC. All modalities are corrected for multiple comparisons (*P*<0.05). Crosshair is at −1/−56/30 mm MNI-space.

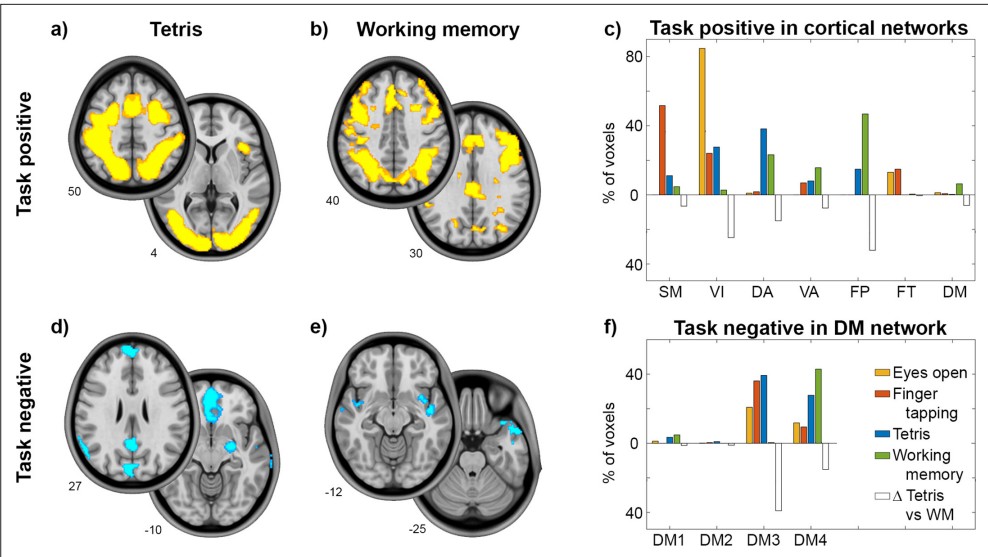

**Figure 2.** Detailed response for the cognitive tasks. Task effects represent the overlap computed as the intersection between BOLD signal changes and glucose metabolism (all *P*<0.05 corrected). Slices show major clusters of positive and negative responses for the two tasks and numbers indicate the z-axis in mm MNI space. Bar graphs show the percentage of voxels with a positive task response for each of the 7 cortical networks (*Yeo et al., 2011*). (**a–c**) Differences in positive task responses between Tetris (DS1) and working memory (DS2) (*Stiernman et al., 2021*) were most pronounced in visual, dorsal attention and fronto-parietal networks, as indicated by open bars (absolute difference between Tetris and working memory). For completeness, previous CMRGlu data obtained while opening the eyes (orange) and right finger tapping (red) was also included (DS3) (*Hahn et al., 2018*). These elicited the main task-positive response in visual and somato-motor networks, respectively. (**d-f**) Negative task responses are shown for DMN subparts as given by the 17-network parcellation (*Yeo et al., 2011*), with DMN3 and DMN4 covering mostly core (PCC, mPFC, angular) and ventral areas (temporal, lateral OFC, superior frontal), respectively. The negative response was strongest in DMN3 for Tetris, visual and motor tasks, but in DMN4 for the working memory task. The number of voxels per network was normalized by the total number of activated (**c**) or deactivated (**f**) voxels across both imaging modalities. Thus, each task sums up to 100% across all cortical networks.

The online version of this article includes the following figure supplement(s) for figure 2:

**Figure supplement 1.** Task response computed by statistical conjunction analysis.

(DS1), we evaluated the spatial overlap of negative task responses in the cerebral metabolic rate of glucose (CMRGlu quantified with the Patlak plot) and the BOLD signal specifically in the pmDMN. Next, comparisons to other tasks were drawn, focusing on the corresponding positive task effects across different large-scale functional networks. For this purpose, group-average statistical results of previously published data sets were re-analyzed. These comprised the aforementioned working memory task, specifically the difficult manipulation condition which required active continuation of alphabetic letters (DS2) (*Stiernman et al., 2021*), as well as data from simple eyes open and finger tapping conditions (DS3; *Hahn et al., 2018*). After that, the distinct spatial activation patterns across different tasks were used to quantitatively characterize the CMRGlu response of the pmDMN in DS1. Finally, we investigated the directional influence between the pmDMN and task-positive networks using metabolic connectivity mapping (MCM; *Riedl et al., 2016*).

## Regional and task specific effects of CMRGlu and BOLD changes

We first assessed the regional overlap of task-induced changes in CMRGlu and the BOLD signal for the Tetris paradigm specifically for the DMN. This was directly compared to previous results from the working memory task. The Tetris task elicited consistent negative responses for both the BOLD signal and CMRGlu in midline core regions of the DMN, such as the medial prefrontal cortex (mPFC) and the PCC/precuneus (DS1, *Figure 1a*, all p<0.05 FWE corrected at cluster level, high threshold of p<0.001 uncorrected). This included a cluster in the pmDMN, previously referred to as the ventral PCC (*Leech et al., 2011*). In contrast, working memory was associated with a dissociation in the DMN. Here, a

negative BOLD response was accompanied by increased glucose metabolism in the anterior/dorsal part of the PCC (*Figure 1b*; *Stiernman et al., 2021*).

We then compared task-induced changes across all task paradigms, evaluating relationships between task-positive and task-negative effects. Using a common cortical parcellation scheme of seven functional networks (*Yeo et al., 2011*), this analysis revealed distinct spatial patterns for the different tasks (*Figure 2*). Tetris elicited positive changes for BOLD and CMRGlu predominantly in the visual network (VIN) and DAN, while working memory mostly involved the FPN. The greatest difference in positive task effects between Tetris and working memory was therefore observed for VIN, DAN, and FPN (white bars in *Figure 2c*), which were selected for further evaluation. Overlapping negative responses for both imaging modalities occurred in DMN3 ('core') and DMN4 ('ventral') for Tetris, but only in DMN4 for working memory (*Figure 2f*). Simple visual stimulation (eyes open vs. eyes closed) and right finger tapping elicited increased CMRGlu in VIN and SMN (somato-motor), respectively, and a negative response mostly in DMN3 (*Figure 2*). Notably, some of the regions with negative responses are particularly prone to susceptibility artifacts in fMRI. Since this issue is not present in fPET, these deactivations do not seem to be solely driven by artifacts. These results were reproduced when computing the overlap between imaging modalities by statistical conjunction analysis in SPM12 (*Figure 2—figure supplement 1*). Thus, the above results were obtained on the basis of common task-specific changes between the BOLD signal and CMRGlu. For completeness, we also evaluated the overall regional agreement between the two imaging modalities, confirming that activations (Dice coefficient Tetris = 0.57 and working memory = 0.35) showed higher overlap than deactivations (Tetris = 0.16 and working memory = 0.06; *Stiernman et al., 2021*).

## Task positive CMRGlu shapes pmDMN response

We next characterized the negative CMRGlu response in the pmDMN observed in DS1. Individual CMRGlu values were extracted for this region and from the networks that showed the greatest difference between Tetris and the working memory tasks (VIN, DAN, and FPN). These were then entered in a linear multiple regression analysis.

The negative CMRGlu task response in the pmDMN during Tetris significantly covaried with the positive CMRGlu response in the other networks (*Figure 3a*, F = 4.84, p = 0.005). More specifically, CMRGlu of the pmDMN was associated with that of FPN (b = 0.75, p = 0.006) and DAN (b = −0.68, p = 0.010), but not VIN (b = 0.30, p > 0.19). Notably, the influence of DAN and FPN was in the opposite direction (as given by the opposite sign of the parameter estimates). That is, across individuals, the combination of low glucose metabolism in FPN and high metabolism in DAN was associated

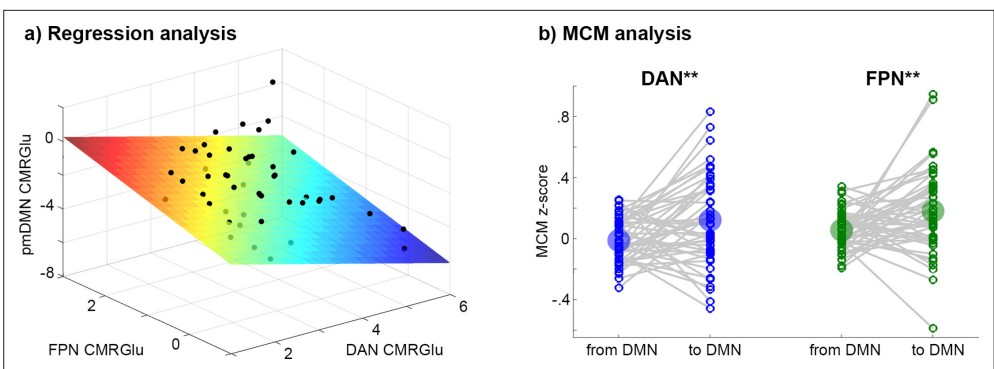

**Figure 3.** Relationship of CMRGlu response between networks. (**a**) Visualization of regression analysis results for the Tetris task (DS1, *F*=4.84, p=0.005). Positive CMRGlu task responses of the FPN (p=0.006) and the DAN (p=0.010) both explained the negative CMRGlu response in the pmDMN across subjects. Here, FPN and DAN exerted an inverse association with pmDMN, where low glucose metabolism in FPN and high metabolism in DAN yield a negative CMRGlu response in the pmDMN. Units on all axes are µmol/100 g/min. (**b**) MCM analysis for the Tetris task (DS1) combining the association of functional connectivity and CMRGlu for causal inference on directionality. The influence from DAN and FPN to DMN was significantly stronger than vice versa (both **p<0.01). The same regions were used for regression and MCM analyses: For FPN and DAN, CMRGlu was extracted from voxels showing a significant overlap of activations between imaging modalities but no overlap between Tetris and WM tasks.

with a negative CMRGlu task response in the pmDMN. Similar results were obtained when defining the overlap across imaging modalities as a formal statistical conjunction (whole model: F = 3.32, p = 0.028; FPN: b = 0.39, p = 0.018, DAN: b = −0.55, p = 0.011, VIN: b = 0.17, p = 0.33). The association was also observed when using the atlas-based network definition (*Yeo et al., 2011*), that is, without incorporating prior knowledge of positive task responses (whole model: F = 6.65, p = 0.0008, FPN: b = 1.04, p = 0.002, DAN: b = −0.98, p = 0.001, VIN: b = 0.69, p = 0.030). The influence of the visual network also reached significance when using the atlas definition, in line with our initial work showing CMRGlu decreases in the pmDMN when comparing eyes opened vs. eyes closed (*Hahn et al., 2018*).

Finally, MCM was employed to assess the putative direction of influence between task-positive networks and the DMN. The technique is based on previous evidence showing that most energy demands emerge postsynaptically (*Harris et al., 2012*; *Mergenthaler et al., 2013*; *Yu et al., 2018*), which is used to identify the target region of a connection. Applied to imaging parameters, this is reflected in a correlation of spatial patterns between functional connectivity and the underlying CMRGlu, thereby enabling causal inferences (see Methods). In line with the regression analysis, MCM indicated that during task performance the direction of influence is from DAN (MCM z-score = 0.12 to −0.01, p = 0.005, *Figure 3b*) and FPN (MCM = 0.18 vs. 0.05, p = 0.006) to the pmDMN.

## Discussion

Using simultaneous PET/MR imaging, we find that the metabolic demands linked to pmDMN BOLD deactivations depend on the actual task at hand and the correspondingly activated functional networks (visual/motor and attention vs. control). Specifically, we show that high task-induced metabolism in the DAN but low metabolism in the FPN lead to a negative CMRGlu response in the pmDMN. These findings resolve the discrepancy between (non)congruent glucose demands and BOLD signal changes across different cognitive tasks, emphasizing the distinct metabolic underpinnings of BOLD signal deactivations during cognitive processing.

### Spatially distinct metabolic response within the DMN

The tasks investigated in this work resulted in distinct metabolic responses in the pmDMN (*Figures 1–2*). This included the PCC, a spatially and functionally heterogeneous region (*Leech et al., 2014*) with ventral and dorsal subcomponents (vPCC/dPCC) (*Vogt et al., 2006*). The PCC is a central hub, connecting brain networks involved in complex behavior (*Hagmann et al., 2008*). Specifically, the PCC is thought to act as an interface between distributed functional networks by echoing their activity (*Leech et al., 2011*) and tuning the balance between the internal and external focus of attention (*Leech and Sharp, 2014*). Despite a consistent negative BOLD response during external tasks, the dPCC and vPCC are differently integrated into the DMN and distinct large-scale networks underpinning task performance (*Leech et al., 2011*; *Parker and Razlighi, 2019*). That is, these two DMN regions are thought to be flexibly recruited to support various cognitive functions (*Cocchi et al., 2013*).

The vPCC is engaged in tasks with an internal focus of attention (*Leech and Sharp, 2014*), such as autobiographical memory recollection (*Svoboda et al., 2006*), and when demands for externally directed attention are low (*Leech et al., 2011*). Our findings add to this knowledge by suggesting that vPCC BOLD deactivation represents a downregulation of internal processing in favor of a focus on the external task for example, as supported by DAN, VIN, and SMN (*Leech et al., 2011*). Thus, a negative BOLD response paralleled by a decreased CMRGlu, as observed for Tetris as well as simple visual and motor stimulation, indicates a reduction of overall metabolism, in line with the original understanding of DMN task-negativity (*Fox et al., 2005*; *Hayden et al., 2009*; *Raichle et al., 2001*).

In contrast, the dPCC is involved in externally directed attention and plays an opposite role to the vPCC (*Leech et al., 2011*). With increasing working memory load, the dorsal subregion exhibits stronger integration with the DMN and more pronounced BOLD signal anticorrelation with task-control networks, which is also reflected in behavioral performance (*Leech et al., 2011*). This could lead to a competition between self-generated and task-induced demands. Thus, vPCC BOLD deactivation during working memory seems to reflect a metabolically expensive process that suppresses self-generated thoughts to enhance task focus, particularly when internal task demands are high (*Buckner and DiNicola, 2019*; *Hearne et al., 2015*; *Leech et al., 2011*; *Stiernman et al., 2021*).

## Metabolic and neurophysiological considerations

The distinct relationships between BOLD and CMRGlu signals that emerge during specific tasks highlight the different physiological processes contributing to neuronal activation of cognitive processing (*Goyal and Snyder, 2021*; *Singh, 2012*). While CMRGlu measured by fPET provides an absolute indicator for glucose consumption, the BOLD signal reflects deoxyhemoglobin concentration, which depends on various factors, such as cerebral blood flow (CBF), cerebral blood volume (CBV), and the cerebral metabolic rate of oxygen (CMRO$_2$; *Goense et al., 2016*). In simple terms, the BOLD signal relates to the ratio of $\Delta$CBF/$\Delta$CMRO$_2$.

Assuming that the observed BOLD decreases during Tetris and WM emerge from the same mechanisms, this would result in a comparable $\Delta$CBF/$\Delta$CMRO$_2$ in the pmDMN for both tasks. Given that these types of tasks (external attention and cognitive control) elicit a reduction in CBF in the pmDMN (*Shulman et al., 1997*; *Zou et al., 2011*), CMRO$_2$ also decreases albeit to a lesser extent (*Raichle et al., 2001*). Therefore, the respective metabolic processes can be described by their oxygen-to-glucose index (OGI), the ratio of CMRO$_2$/CMRGlu. Accordingly, our results suggest two distinct pathways underlying BOLD deactivations in the pmDMN that differ regarding their OGI. During Tetris there is a BOLD deactivation with a high OGI, resulting from a larger decrease in CMRGlu than CMRO$_2$. This metabolically inactive state is in line with electrophysiological recordings in humans (*Fox et al., 2018*) and in non-human primates showing a decrease of neuronal activity in the pmDMN that covaries with the degree of exteroceptive vigilance (*Bentley et al., 2016*; *Hayden et al., 2009*; *Shmuel et al., 2006*). Therefore, we suggest that the negative BOLD response during external tasks reflects a reduction of neuronal activity and their respective metabolic demands. On the other hand, the relatively increased CMRGlu without the corresponding surge in CMRO$_2$ (combined with decreased CBF) hints at another kind of BOLD deactivation with a low OGI in the pmDMN during working memory, indicating energy supply by aerobic glycolysis (*Blazey et al., 2019*; *Vaishnavi et al., 2010*). Previous work in non-human primates has indeed suggested a differential coupling of neuronal activity to hemodynamic oxygen supply in this region (*Bentley et al., 2016*). Furthermore, tonic suppression of PCC neuronal spiking during task performance was punctuated by positive phasic responses (*Hayden et al., 2009*), which could indicate differences between both tasks also at the level of electrophysiologically measured activity.

On the neurotransmitter level, one of the current hypotheses regarding BOLD deactivations proposes that CMRO$_2$ and CBF are affected by the balance of the excitatory and inhibitory neurotransmitters, specifically GABA and glutamate (*Buzsáki et al., 2007*; *Lauritzen et al., 2012*; *Sten et al., 2017*). In the PCC, glutamate release prevents negative BOLD responses (*Hu et al., 2013*), whereas a lower glutamate/GABA ratio is associated with greater deactivation (*Gu et al., 2019*). As glutamate elicits proportional glucose consumption (*Lundgaard et al., 2015*; *Zimmer et al., 2017*), decreases in glutamate signaling in the pmDMN could indeed explain both, the decreased BOLD response and decreased CMRGlu during the Tetris task. Conversely, increased GABA supports a negative BOLD response in the PCC (*Hu et al., 2013*), as do working memory tasks (*Koush et al., 2021*) and pharmacological stimulation with GABAergic benzodiazepines (*Walter et al., 2016*). In consequence, the observed dissociation between BOLD changes and CMRGlu during working memory could indeed result from metabolically expensive (*Harris et al., 2012*) GABAergic suppression of the BOLD signal (*Stiernman et al., 2021*). However, we need to emphasize that glutamate and GABAergic signaling was not measured in the current study, thus, the above interpretations are of speculative nature. Nonetheless, future work may test this promising hypothesis, for example, using pharmacological alteration of GABAergic and glutamatergic signaling or optogenetic approaches modulating GABAergic interneuron activity.

## Limitations, outlook, and conclusions

To summarize, our work provides novel insights into the metabolic underpinnings of negative BOLD responses in the pmDMN, showing that regionally specific effects depend on the functional networks involved in task execution. Acknowledging the underlying energy demands and neurotransmitter actions underpinning neuronal activity is necessary to understand PCC function, including how this essential DMN region dynamically interacts with macroscale networks as a function of changing behavioral demands (*Kelly et al., 2008*; *Hampson et al., 2006*). While our work provides valuable information to address this knowledge gap, several caveats are worth noting: First, we did not assess

the different tasks in the same cohorts, but pooled different studies. However, only effects corrected at the group level were used as obtained from commonly employed sample sizes, which should provide representative findings. Second, Tetris and WM data were acquired with different acquisition details and task designs, that is, continuous task performance versus hierarchical embedding of short task blocks for BOLD into longer PET acquisition, respectively. As the latter may not clearly differentiate between start-cue and task activation, this may limit transferability. Therefore, future studies investigating these effects should address this limitation, ideally studying the different tasks in the same cohort, with a comparable task design. Third, our results were obtained using rest with crosshair fixation as the baseline condition. Since task-specific effects in the BOLD signal and CMRGlu are relative to this baseline condition, these would likely change if using an active control condition as baseline. Fourth, additional contextual load input to the pmDMN, (e.g., task-relevant emotional content) may be another important factor affecting its activation (*Fan et al., 2019*). Future studies may therefore include additional cognitive and emotional domains. Of particular interest would be the investigation of introspective tasks such as autobiographical memory as these typically induce a positive BOLD response in the PCC, while the coupling with the CMRGlu response is unknown. Such paradigms would also allow to assess whether the presently observed network interactions are symmetrical, that is, if task positive networks show decreased activation when the DMN exhibits a positive response. This hypothesis seems reasonable in the light of recent work reporting bidirectional information exchange between default mode and other networks (*Das et al., 2022*; *Menon et al., 2023*). Although the DMN, and in particular the PCC, has been implicated in numerous brain disorders (*Buckner and DiNicola, 2019*), our data suggests that this could be mediated by other interacting cortical networks. Assessing the differential influence of attention and control networks on the pmDMN may therefore represent an interesting approach to improve our understanding of network dysfunction in different patient populations.

## Methods

Throughout this manuscript, we refer to deactivation/decrease/negative response and likewise to activation/increase/positive response as relative changes compared to the baseline condition. That is, changes in the BOLD signal and glucose metabolism (here as obtained from general linear model analyses) emerge from a negative or positive contrast sign with respect to the baseline. For glucose metabolism, these changes are absolutely quantified in µmol/100 g/min with the arterial input function and the Patlak plot.

In this work, the baseline condition for all tasks was a resting-state defined as looking at a crosshair without focusing on anything in particular, except for the eyes open condition which used eyes closed as baseline (please see discussion for implications when other control conditions are employed).

### Data sets

The primary dataset used in this work (DS1) consists of simultaneously acquired fPET/fMRI data from n=50 healthy subjects during the performance of a challenging visuo-motor task (i.e., the video game Tetris). Unless stated otherwise, all data, methods and results refer to DS1, including data of individual participants and all statistics across subjects. A detailed description of the study design, cognitive task, fPET/fMRI measurements and first-level analysis of DS1 is given in our previous work (*Hahn et al., 2020*) and also below.

For direct comparison with previous work, two further data sets were used in this study. These only include group-averaged results with contrasts and statistical maps as published previously. Dataset 2 (DS2) consists of fPET/fMRI data from n=23 healthy subjects (mean age ± sd = 25.2±4.0 years, 13 females) acquired during the performance of a working memory manipulation task, which required active transformation of stimuli in working memory (*Stiernman et al., 2021*). Group-level maps of significant increases and decreases in [18F]FDG glucose metabolism and the BOLD signal during task execution as compared to baseline were used (p<0.05 TFCE corrected). Dataset 3 (DS3) comprises data on glucose metabolism obtained from [18F]FDG fPET imaging (*Hahn et al., 2018*). We included group-average statistical maps from n=18 healthy subjects (24.2±4.3 years, 8 females), who performed cognitively simple tasks of eyes open vs. eyes closed and during right finger tapping vs. rest (p<0.05 FWE corrected cluster-level after P<0.001 uncorrected voxel level).

For all datasets and tasks, the baseline condition was defined as looking at a crosshair at resting-state, except for the eyes open condition of DS3 which used eyes closed as the baseline.

## Experimental design

All participants of DS1 underwent one PET/MRI scan while performing a challenging visuo-spatial cognitive task. Data acquisition started with structural imaging (8 min). This was followed by 52 min fPET, which comprised an 8 min baseline at rest and then four periods of continuous task performance (6 min each, two easy and two hard conditions, randomized) with periods of rest after each task block (5 min). Simultaneously with fPET, BOLD fMRI was acquired during the continuous task execution (6 min each), which was used for the calculation of metabolic connectivity mapping. Finally and immediately after fPET, another BOLD fMRI sequence was obtained in a conventional block design with the same task (12 task blocks, 30 s each, four easy, four hard and four control blocks, 10 s baseline between task blocks, 8.17 min in total). This acquisition was used for the computation of BOLD-derived neuronal activation. Further acquisitions (diffusion weighted imaging, BOLD imaging at rest, arterial spin labelling) were not used in the current work. During all periods of rest, participants were instructed to look at a crosshair, relax and not to focus on anything in particular.

## Cognitive task

An adapted version of the video game Tetris (https://github.com/jakesgordon/javascript-tetris MIT license; *Jake and Richard, 2011*) was implemented in electron 1.3.14. The aim is to build complete horizontal lines by rotation and alignment of bricks, which descend from the top of the screen. The task included two levels of difficulty, which differed regarding the speed of the descending bricks (easy/hard: 1/3 lines per sec) and the number of incomplete lines built at the bottom (easy/hard: 2/6 lines out of 20). The control condition of the BOLD acquisition was not used in this work. Right before the start of the PET/MRI scan, participants familiarized themselves with the control buttons by 30 s training of each task condition. The employed task represents a cognitively challenging paradigm, which requires a high level of attention, rapid visuo-spatial motor coordination, mental rotation, spatial planning and problem solving.

## Participants

Fifty-three healthy participants were initially recruited for DS1 and 50 were included in the current analysis (mean age ± sd = 23.3±3.4 years, 23 females). Reasons for drop out were failure of arterial blood sampling (n=1) and technical issues during the scan (n=2). In part these subjects also participated in previous studies (*Hahn et al., 2020*; *Klug et al., 2022*; *Rischka et al., 2021*). Previous work using fPET demonstrated robust changes in glucose metabolism during cognitive processing with sample sizes between 10 and 23 participants (*Hahn et al., 2020*; *Jamadar et al., 2019*; *Stiernman et al., 2021*). As the sample of the current study is at least twofold larger, a former sample size estimation was omitted. All subjects completed an initial screening to ensure general health through a routine medical examination (blood tests, electrocardiography, neurological testing, structural clinical interview for DSM-IV). Female participants also underwent a urine pregnancy test at the screening visit and before the PET/MRI examination. Exclusion criteria were current or previous somatic, neurological or psychiatric disorders (12 months), substance abuse or psychopharmacological medication (6 months), current pregnancy or breast feeding, contraindications for MRI scanning, previous study-related radiation exposure (10 years) and previous experience with the video game Tetris (3 years). All participants provided written informed consent after a detailed explanation of the study protocol, they were insured and reimbursed for participation. The study was approved by the Ethics Committee of the Medical University of Vienna (ethics number 1479/2015) and procedures were carried out according to the Declaration of Helsinki. The study was pre-registered at ClinicalTrials.gov (NCT03485066).

## PET/MRI data acquisition

Participants had to fast for at least 5.5 hr before the start of the PET/MRI scan, except for unsweetened water. The radiotracer [18F]FDG was applied in a bolus +infusion protocol (510 kBq/kg/frame for 1 min, 40 kBq/kg/frame for 51 min) using a perfusion pump (Syramed μSP6000, Arcomed, Regensdorf, Switzerland), which was kept in an MRI-shield (UniQUE, Arcomed).

MRI acquisitions included a T1-weighted structural scan (MPRAGE sequence, TE/TR = 4.21/2200ms, TI = 900ms, flip angle = 9°, matrix size = 240 x 256, 160 slices, voxel size = 1 x 1 x 1 mm +0.1 mm gap, 7.72 min) and BOLD fMRI (EPI sequence, TE/TR = 30/2000ms, flip angle = 90°, matrix size = 80 x 80, 34 slices, voxel size = 2.5 x 2.5 x 2.5 mm +0.825 mm gap, 6 min for functional connectivity and 8.17 min for neuronal activation in the block design).

## Data acquisition of DS2 (working memory task)

Data were obtained as described previously (*Stiernman et al., 2021*). Briefly, participants fasted for 4 hr before the scan. Intravenous infusion of 180 MBq [18F]FDG was started simultaneously with PET/MRI acquisition (GE Signa) and lasted 60 min (0.016 ml/s). MRI scans included an attenuation correction sequence, T1-weighted structural imaging (TE/TR = 3.1/7200ms, flip angle = 12°, matrix size = 256 x 256, 180 slices, voxel size = 0.49 x 0.49 x 1, mm, 7.36 min) and BOLD functional MRI (EPI sequence, TE/TR = 30/4000ms, flip angle = 80°, matrix size = 96 x 96, voxel size = 1.95 x 1.95 x 3.9 mm, 42 min).

During PET/MRI acquisition participants kept their eyes open. The working memory task was completed in a hierarchical design. That is, short task blocks of 45 s and 15 s rest served for assessment of task changes in the BOLD signal. These short blocks were embedded in long 6 min blocks, which enabled identification of task effects in glucose metabolism. Participants completed 2x6 min maintenance and 2x6 min manipulation blocks, with 3x6 min rest blocks in-between. In the maintenance condition, 4 target letters were shown and participants were asked if a delayed probe letter matches one of the targets. In the manipulation condition, 2 target letters were shown and participants were required to indicate if a delayed probe letter represents the subsequent letter in the alphabet of any of the targets.

PET data were reconstructed to 60x1 min frames and analyzed with the general linear model (GLM). Since arterial blood samples were not available for DS2, beta estimates from the GLM were entered into group-level statistical analysis (p<0.05 TFCE corrected). The contrast of interest was manipulation vs. rest. Confirmatory kinetic modeling was performed with a literature-based arterial input function.

## Data acquisition of DS3 (eyes open and finger tapping tasks)

Data were obtained as described previously (*Hahn et al., 2018*). In short, participants fasted for at least 5.5 hr before the scan. Intravenous infusion of [18F]FDG (3 MBq/kg body weight) was started simultaneously with PET/MRI acquisition (Siemens mMR) and lasted 95 min (36 ml/h). MRI scans included T1-weighted structural imaging (MPRAGE, TE/TR = 4.2/2000ms, TI = 900ms, flip angle = 9°, matrix size = 256 x 240, 160 slices, voxel size = 1 x 1 x 1 mm +0.1 mm gap, 7.02 min) and BOLD functional MRI (EPI sequence, TE/TR = 30/2440ms, flip angle = 90°, matrix size = 100 x 100, 30 slices, voxel size = 2.1 x 2.1 x 3 mm +0.75 mm gap, 5 min per task block).

During PET/MRI acquisition participants kept their eyes closed. Continuous task performance included opening the eyes for 2x10 min and 2x10 min tapping the right thumb to the fingers with 15 min rest blocks in-between. Arterial blood samples were obtained during the rest periods.

PET data were reconstructed to 95x1 min frames and analyzed with the GLM. Quantification of CMRGlu was carried out with the arterial input function and the Patlak plot. CMRGlu maps were entered into group-level statistical analysis (p<0.05 FWE corrected cluster-level after p<0.001 uncorrected voxel level). The contrast of interest was eyes open vs. eyes closed or finger tapping vs. rest.

## Blood sampling

Before the PET/MRI scan blood glucose levels were assessed as triplicate (Glu$_{plasma}$). During the PET/MRI acquisitions, manual arterial blood samples were drawn at 3, 4, 5, 14, 25, 36, and 47 min after the start of the radiotracer administration (*Rischka et al., 2018*). From these samples, whole-blood and plasma activity were measured in a gamma counter (Wizard$^2$, Perkin Elmer). The arterial input function was obtained by linear interpolation of the manual samples to match PET frames and multiplication with the average plasma-to-whole-blood ratio.

## Cerebral metabolic rate of glucose metabolism (CMRGlu)

PET images were reconstructed and processed as described previously (*Rischka et al., 2018*). Briefly, PET list mode data were corrected for attenuation with a database approach (*Burgos et al., 2014*)

and reconstructed to 30 s frames (matrix size = 344 x 344, 127 slices). Preprocessing was carried out in SPM12 (https://www.fil.ion.ucl.ac.uk/spm/) and comprised motion correction (quality = 1, register to mean), spatial normalization to MNI space using the T1-weighted structural image and spatial smoothing with an 8 mm Gaussian kernel. Non-gray matter voxels were masked out and a low pass filter was applied, which induces temporal smoothing (cutoff frequency = 3 min). The rationale for this filter is to reduce noise in the high temporal resolution [$^{18}$F]FDG signal. Since task blocks lasted 6 min (or even longer for DS3), we assume that changes faster than this do not reflect task-related effects. Identification of task-specific effects was done with the general linear model. Four regressors were included to characterize the baseline, the two task conditions (easy and hard as a linear function with slope = 1 kBq/frame) and head motion (the first principal component of the 6 motion parameters). The baseline regressor was given by the average time course of all gray matter voxels, excluding those activated during the hard task condition of the individual BOLD block design (p<0.05 FWE corrected voxel level). This approach provides the best model fits (*Rischka et al., 2018*) without negatively affecting task-specific changes in glucose metabolism (*Hahn et al., 2020*; *Rischka et al., 2018*) or test-retest reliability (*Rischka et al., 2021*). Quantification was carried out with the Patlak plot (t* fixed to 15 min) and the influx constant $K_i$ was converted to CMRGlu as

$$CMRGlu = K_i * Glu_{plasma}/LC * 100$$

with LC being the lumped constant = 0.89 (*Graham et al., 2002*; *Wienhard, 2002*). The resulting maps represented individual task-specific changes in CMRGlu, which were used for group-level analyses and MCM.

## Blood oxygen level dependent (BOLD) signal changes

BOLD imaging data of the block design were processed with SPM12 as described previously (*Rischka et al., 2018*). In short, data were corrected for slice timing effects (reference = middle slice) and head motion (quality = 1, register to mean), normalized to MNI-space via the T1-weighted image and smoothed with an 8 mm Gaussian kernel. First-level analysis was carried out to assess individual estimates of the BOLD task response. Here, regressors were included for the two task conditions (easy, hard) and the control condition as well as nuisance regressors for head motion, white matter and CSF signals. The contrast of interest was chosen as the hard task level vs. baseline to facilitate comparison to previous work (*Stiernman et al., 2021*).

## Metabolic connectivity mapping (MCM)

MCM was calculated during the performance of the hard task between regions of interest as published previously (*Hahn et al., 2020*). The approach combines patterns of functional connectivity (FC) and CMRGlu and enables to estimate directional connectivity between brain regions (*Riedl et al., 2016*). MCM relies on physiological characteristics of energy demands, which mostly emerge postsynaptically (*Harris et al., 2012*; *Mergenthaler et al., 2013*) and thereby allow to identify the target region of a connection. Briefly, computing the FC between region A and region B yields a distinct FC voxel-wise pattern in region B. If the influence of region A on B is causal, this will result in a corresponding CMRGlu pattern in region B because of the coupling between the BOLD signal and the underlying glucose metabolism (*Attwell et al., 2010*; *Mishra et al., 2016*). FC was computed from continuously acquired BOLD imaging data during the performance of the hard task condition. Preprocessing was done as described above for the BOLD block design. After spatial smoothing, motion scrubbing was performed by removing frames with a displacement >0.5 mm (plus one frame back and two forward). To remove potentially confounding signals linear regression was used (including motion parameters, white matter and cerebrospinal fluid signals), followed by band-pass filtering (0.01<f < 0.15 Hz *Sun et al., 2004*). FC was then calculated as the temporal correlation between regions A and B. MCM was computed as the spatial correlation between voxel-wise patterns of FC and CMRGlu in region B (both correlations were z-transformed). ROIs were defined as for the regression analysis below, i.e., as voxels of the DAN and FPN which were specific for each task as well as voxels of the pmDMN identified with DS1. One subject was excluded from MCM analysis because of movement during the BOLD acquisition of the hard task condition.

## Statistical analysis

For DS1, task responses in CMRGlu and BOLD were evaluated during the Tetris task with separate one sample t-tests for each imaging modality (p<0.05 FWE corrected cluster-level after p<0.001 uncorrected voxel level). Group maps with significant changes were then binarized and the overlap between CMRGlu and BOLD responses was computed as their intersection. First, this was done for negative task effects to assess if DMN deactivations are present in both modalities.

Next, we tested the hypothesis that negative responses in the DMN are mirrored by the corresponding positive task changes that are specific to other cortical networks. To provide a comprehensive overview of changes across different tasks, we also included results from a working memory task (DS2; *Stiernman et al., 2021*) as well as eyes opened vs. eyes closed conditions and right finger tapping (DS3; *Hahn et al., 2018*). For the Tetris and working memory tasks, overlapping increases in CMRGlu and BOLD signal changes were computed in the same way as above, that is, by computing the intersection of group-level significant positive task effects across modalities for each task separately. For the visual and motor paradigms, only CMRGlu data was available.

The different positive task responses were first compared qualitatively by computing the percentage of activated voxels for each task and 7 cortical networks (*Yeo et al., 2011*). Negative task responses were calculated at a finer level of detail with the DMN subdivisions obtained from the 17 network parcellation (*Yeo et al., 2011*).

After that, these network-specific differences in positive responses between the Tetris and working memory tasks (i.e., DS1 vs. DS2) were then used to quantitatively explain the CMRGlu decreases in the pmDMN observed in the current dataset. Individual CMRGlu values of DS1 were extracted for visual, dorsal attention and fronto-parietal networks, as these networks showed greatest differences in activation (see results). Here, only voxels were used which showed an overlap of significant task changes (i.e., the intersection between CMRGlu and BOLD signal) and which were specific for each task (i.e., voxels that did not overlap between the two tasks). Voxels of the pmDMN were defined by the overlapping negative task response between CMRGlu and BOLD changes of DS1. The extracted values were then entered into a regression analysis to characterize the CMRGlu response in the pmDMN. To assess the robustness of our results and allow for generalization, two control analyses were performed. First, the overlap of task responses across imaging modalities at the group-level was also computed by a formal conjunction analysis in SPM12 for the Tetris and working memory tasks. That is, for each task the individual maps of CMRGlu and BOLD response were z-scored and entered in a one-way ANOVA with each modality representing a 'group'. The separate contrasts for each of the modalities were then combined in a conjunction (p < 0.05 FWE corrected cluster level following p < 0.001 uncorrected voxel level). Second, the regression was also calculated when extracting CMRGlu values from the entire visual, dorsal attention and fronto-parietal networks (*Yeo et al., 2011*), i.e., independent of regionally specific prior knowledge on positive task responses.

Finally, we assessed the directionality of the association between DMN and task-positive networks in DS1. Individual MCM values were entered in a paired t-test to assess differences between directions, for example, DAN->DMN vs. DMN->DAN.

## Acknowledgements

We thank the graduated team members and the diploma students of the Neuroimaging Lab (NIL, head: R Lanzenberger) as well as the clinical colleagues from the Department of Psychiatry and Psychotherapy for clinical and/or administrative support. In detail, we would like to thank S Kasper, K Papageorgiou, P Michenthaler, T Vanicek, A Basaran, M Hienert, L Silberbauer, J Unterholzner and G Gryglewski for medical support, L Rischka and M B Reed for analysis support, V Ritter, K Einenkel and E Sittenberger for subject recruitment and A Jelicic for partly implementation of the task. We are further grateful to J Völkle and A Pomberger for radioligand synthesis. The scientific project was performed with the support of the Medical Imaging Cluster of the Medical University of Vienna.

This research was funded in whole, or in part, by the Austrian Science Fund (FWF) KLI 610, PI: A Hahn. For the purpose of open access, the author has applied a CC BY public copyright license to any Author Accepted Manuscript version arising from this submission. S Klug is supported by the MDPhD Excellence Program of the Medical University of Vienna. A Rieckmann and L Stiernman are supported by the European Research Council under the European Union's Horizon 2020 research

and innovation program ERC-STG-716065 to A Rieckmann. L Cocchi is supported by the Australian NHMRC (GN2001283).

# Additional information

## Competing interests

Wolfgang Wadsak: declares to having received speaker honoraria from the GE Healthcare and research grants from Ipsen Pharma, Eckert-Ziegler AG, Scintomics, and ITG; and working as a part time employee of CBmed Ltd. (Center for Biomarker Research in Medicine, Graz, Austria). Marcus Hacker: received consulting fees and/or honoraria from Bayer Healthcare BMS, Eli Lilly, EZAG, GE Healthcare, Ipsen, ITM, Janssen, Roche, and Siemens Healthineers. Rupert Lanzenberger: received investigator-initiated research funding from Siemens Healthcare regarding clinical research using PET/MRI. He is a shareholder of the start-up company BM Health GmbH since 2019. The other authors declare that no competing interests exist.

## Funding

| Funder | Grant reference number | Author |
| --- | --- | --- |
| Austrian Science Fund | KLI610 | Andreas Hahn |
| Medical University of Vienna | MDPhD Excellence Programm | Sebastian Klug |
| European Research Council | ERC-STG-716065 | Anna Rieckmann Lars Stiernman |
| National Health and Medical Research Council | GN2001283 | Luca Cocchi |

The funders had no role in study design, data collection and interpretation, or the decision to submit the work for publication.

## Author contributions

Godber M Godbersen, Investigation, Writing – original draft, Writing – review and editing; Sebastian Klug, Verena Pichler, Julia Raitanen, Investigation, Writing – review and editing; Wolfgang Wadsak, Supervision, Investigation, Writing – review and editing; Anna Rieckmann, Luca Cocchi, Michael Breakspear, Methodology, Writing – original draft, Writing – review and editing; Lars Stiernman, Formal analysis, Methodology, Writing – original draft, Writing – review and editing; Marcus Hacker, Rupert Lanzenberger, Conceptualization, Supervision, Writing – review and editing; Andreas Hahn, Conceptualization, Data curation, Software, Formal analysis, Supervision, Funding acquisition, Investigation, Visualization, Methodology, Writing – original draft, Project administration, Writing – review and editing

## Author ORCIDs

Godber M Godbersen http://orcid.org/0000-0002-9739-0724
Anna Rieckmann http://orcid.org/0000-0002-5389-1578
Luca Cocchi http://orcid.org/0000-0003-3651-2676
Rupert Lanzenberger http://orcid.org/0000-0003-4641-9539
Andreas Hahn http://orcid.org/0000-0001-9727-7580

## Ethics

All participants provided written informed consent after a detailed explanation of the study protocol, they were insured and reimbursed for participation. The study was approved by the Ethics Committee of the Medical University of Vienna (ethics number 1479/2015) and procedures were carried out according to the Declaration of Helsinki. The study was pre-registered at ClinicalTrials.gov (NCT03485066).

## Decision letter and Author response

Decision letter https://doi.org/10.7554/eLife.84683.sa1

Author response https://doi.org/10.7554/eLife.84683.sa2

## Additional files

### Supplementary files
• MDAR checklist

### Data availability
Raw data will not be publicly available due to reasons of data protection. Sharing of raw data requires a data sharing agreement, approved by the departments of legal affairs and data clearing of the Medical University of Vienna. Details about this process can be obtained from the corresponding author. Processed data are available at Dryad https://doi.org/10.5061/dryad.5qfttdzbd. Custom code is available at GitHub https://github.com/NeuroimagingLabsMUV/Godbersen2023_eLife, (copy archived at *NeuroimagingLabsMUV, 2023*).

The following dataset was generated:

| Author(s) | Year | Dataset title | Dataset URL | Database and Identifier |
|---|---|---|---|---|
| Godbersen GM, Klug S, Wadsak W, Pichler V, Raitanen J, Rieckmann A, Stiernman L, Cocchi L, Breakspear M, Hacker M, Lanzenberger R, Hahn A | 2023 | Data from: Task-evoked metabolic demands of the posteromedial default mode network are shaped by dorsal attention and frontoparietal control networks | https://doi.org/10.5061/dryad.5qfttdzbd | Dryad Digital Repository, 10.5061/dryad.5qfttdzbd |

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
