## [Editor Report]

This important study advances our understanding of the metabolic and hemodynamic underpinnings of different brain networks. The evidence is convincing, drawn from multiple datasets and including simultaneous fMRI and PET. The study will be of interest to neuroscientists and researchers who use functional neuroimaging tools to study brain activity.

---

## [Decision Letter]

**Decision letter after peer review:**

Thank you for submitting your article "Metabolic demands of the posteromedial default mode network are shaped by dorsal attention and frontoparietal control networks" for consideration by *eLife*. Your article has been reviewed by 3 peer reviewers, and the evaluation has been overseen by a Reviewing Editor and Floris de Lange as the Senior Editor. The following individuals involved in the review of your submission have agreed to reveal their identity: Jonathan Smallwood (Reviewer #2); Tommaso Volpi (Reviewer #3).

Essential revisions:

1) Claims must be tempered to reflect the data. The authors should clearly separate speculation from a conclusion based on the measurements that they made. For example, they measured BOLD/CMRglc and yet they conclude on a GABA mechanism.

2) The distinction between the broader questions, and what the data says, should be made explicit throughout.

3) More details need to be given for the methods so that readers do not have to locate previous papers to understand what was done.

*Reviewer #1 (Recommendations for the authors):*

1. Please clarify what you mean by "We suggest that the former is mediated by decreased glutamate signaling, while the latter results from active GABAergic inhibition" in the Abstract, as it is not clearly evident what is former and latter here.

2. Explain with full clarity and disclosure how the two studies (Tetris and the published working memory, visual and motor stimulations) were different in terms of data acquisition.

*Reviewer #2 (Recommendations for the authors):*

I really enjoyed your study and think it is a great piece of work. I wonder whether there is a boundary condition that could be made more explicit. It may be that the landscape of activity is both context-dependent and symmetrical. It is possible therefore that an analysis of tasks in which the DMN increases activity will lead to a suppression of activity in systems like the DAN. It would be quite easy to modify the language you use to make this clear and to make this a goal for the future (you do mention autobiographical memory in the discussion so could make this point there).

*Reviewer #3 (Recommendations for the authors):*

1) When the authors in the introduction directly talk about "Studies using simultaneous fPET/fMRI" (line 94), I believe fPET and its peculiarities with respect to traditional bolus injection PET, which is the base of most quantitative PET literature, require to be briefly defined since it is not a given that the audience would know this technique.

2) Could the authors elaborate on the negative activations shared by BOLD and [18F]FDG in Figures 2 and S1? Some seem to fall for instance in the temporal pole, i.e., in areas that are known to be prone to susceptibility artifacts in fMRI. Are activations in those areas considered to be reliable?

3) When the anti-correlation between BOLD of task-positive and task-negative networks is brought up, Fox et al. 2005 (https://doi.org/10.1073/pnas.0504136102) might not be the most appropriate reference, as the anticorrelations in that work were largely driven by global signal regression, as later shown by e.g., Murphy et al. 2009 (https://doi.org/10.1016/j.neuroimage.2008.09.036).

4) The application of a low-pass filter applied to the PET time-activity curves (line 477), despite being mentioned in the authors' previous publications, should be briefly justified.

5) When the Patlak plot is employed (line 486), the chosen t-star (t*) should be reported.

6) A reference for the choice of the lumped constant = 0.89 should be provided, as there are controversies over its precise value.

---

## [Author Response]

Essential revisions:1) Claims must be tempered to reflect the data. The authors should clearly separate speculation from a conclusion based on the measurements that they made. For example, they measured BOLD/CMRglc and yet they conclude on a GABA mechanism.

We have revised the manuscript to more clearly separate speculation from direct discussion of our data. Notably, we have added a paragraph that outlines the metabolic underpinnings (glucose vs. oxygen metabolism) of the measured signals. Furthermore, we have limited the discussion on GABA and glutamate and clearly highlight the speculative nature of putative links to neurotransmitter systems.

2) The distinction between the broader questions, and what the data says, should be made explicit throughout.

We have adapted the discussion so that it is now closer to the analyzed data. Thus, in addition to the explicit statements of speculations (response to 1.1, 1.3, 1.4 and 1.5) we have adapted the wording of the discussion to better reflect our observations, which can be found in response to comments 2.1, 2.2, 2.3 and 2.4 as well as 3.0.

3) More details need to be given for the methods so that readers do not have to locate previous papers to understand what was done.

We have included more information on the functional PET method and details on CMRGlu quantification with arterial blood sampling in the manuscript. The methodology is now self-complete.

Reviewer #1 (Recommendations for the authors):1. Please clarify what you mean by "We suggest that the former is mediated by decreased glutamate signaling, while the latter results from active GABAergic inhibition" in the Abstract, as it is not clearly evident what is former and latter here.

We agree with the reviewer that the wording in the abstract is unclear. In addition, interpretations regarding glutamate and GABAergic signaling have been limited throughout the document and clearly labelled as speculative.

Abstract, page 3, line 61:

“While tasks that mainly require an external focus of attention lead to a consistent downregulation of both metabolism and the BOLD signal in the posteromedial DMN, cognitive control during working memory requires a metabolically expensive BOLD suppression. This indicates that two types of BOLD deactivations with different oxygen-to-glucose index may occur in this region. We further speculate that consistent downregulation of the two signals is mediated by decreased glutamate signaling, while divergence may be subject to active GABAergic inhibition.”

2. Explain with full clarity and disclosure how the two studies (Tetris and the published working memory, visual and motor stimulations) were different in terms of data acquisition.

We thank the reviewer for highlighting the difference between the studies. The details about the acquisition schemes of DS2 and DS3 are now also included in the methods section. The limitations already included a statement about these differences and the implications.

Limitations, page 16, line 374:

“Second, Tetris and WM data were acquired with different acquisition details and task designs, i.e., continuous task performance versus hierarchical embedding of short task blocks for BOLD into longer PET acquisition, respectively. As the latter may not clearly differentiate between start-cue and task activation, this may limit transferability. Therefore, future studies investigating these effects should address this limitation, ideally studying the different tasks in the same cohort, with a comparable task design.”

Methods, page 21, line 500:

“Data acquisition of DS2 (working memory task)

Data were obtained as described previously (Stiernman et al. 2021). Briefly, participants fasted for 4 h before the scan. Intravenous infusion of 180 MBq [^18^F]FDG was started simultaneously with PET/MRI acquisition (GE Signa) and lasted 60 min (0.016 ml/s). MRI scans included an attenuation correction sequence, T1-weighted structural imaging (TE/TR = 3.1/7200 ms, flip angle = 12°, matrix size = 256 x 256, 180 slices, voxel size = 0.49 x 0.49 x 1 mm, 7.36 min) and BOLD functional MRI (EPI sequence, TE/TR = 30/4000 ms, flip angle = 80°, matrix size = 96 x 96, voxel size = 1.95 x 1.95 x 3.9 mm, 42 min).

During PET/MRI acquisition participants kept their eyes open. The working memory task was completed in a hierarchical design. That is, short task blocks of 45 s and 15 s rest served for assessment of task changes in the BOLD signal. These short blocks were embedded in long 6 min blocks, which enabled identification of task effects in glucose metabolism. Participants completed 2x 6 min maintenance and 2x 6 min manipulation blocks, with 3x 6 min rest blocks in-between. In the maintenance condition, 4 target letters were shown and participants were asked if a delayed probe letter matches one of the targets. In the manipulation condition, 2 target letters were shown and participants were required to indicate if a delayed probe letter represents the subsequent letter in the alphabet of any of the targets.

PET data were reconstructed to 60x 1 min frames and analyzed with the general linear model (GLM). Since arterial blood samples were not available for DS2, β estimates from the GLM were entered into group-level statistical analysis (p<0.05 TFCE corrected). The contrast of interest was manipulation vs. rest. Confirmatory kinetic modeling was performed with a literature-based arterial input function.”

Methods, page 22, line 523:

“Data acquisition of DS3 (eyes open and finger tapping tasks)

Data were obtained as described previously (Hahn et al. 2018). In short, participants fasted for at least 5.5 h before the scan. Intravenous infusion of [^18^F]FDG (3 MBq/kg body weight) was started simultaneously with PET/MRI acquisition (Siemens mMR) and lasted 95 min (36 ml/h). MRI scans included T1-weighted structural imaging (MPRAGE, TE/TR = 4.2/2000 ms, TI = 900ms, flip angle = 9°, matrix size = 256 x 240, 160 slices, voxel size = 1 x 1 x 1 mm + 0.1 mm gap, 7.02 min) and BOLD functional MRI (EPI sequence, TE/TR = 30/2440 ms, flip angle = 90°, matrix size = 100 x 100, 30 slices, voxel size = 2.1 x 2.1 x 3 mm + 0.75 mm gap, 5 min per task block).

During PET/MRI acquisition participants kept their eyes closed. Continuous task performance included opening the eyes for 2x 10 min and 2x 10 min tapping the right thumb to the fingers with 15 min rest blocks in-between. Arterial blood samples were obtained during the rest periods.

PET data were reconstructed to 95x 1 min frames and analyzed with the GLM. Quantification of CMRGlu was carried out with the arterial input function and the Patlak plot. CMRGlu maps were entered into group-level statistical analysis (p<0.05 FWE corrected cluster-level after p<0.001 uncorrected voxel level). The contrast of interest was eyes open vs. eyes closed or finger tapping vs. rest.”

Reviewer #2 (Recommendations for the authors):I really enjoyed your study and think it is a great piece of work. I wonder whether there is a boundary condition that could be made more explicit. It may be that the landscape of activity is both context-dependent and symmetrical. It is possible therefore that an analysis of tasks in which the DMN increases activity will lead to a suppression of activity in systems like the DAN. It would be quite easy to modify the language you use to make this clear and to make this a goal for the future (you do mention autobiographical memory in the discussion so could make this point there).

We thank the reviewer for bringing up this intriguing idea. Whether these effects are symmetrical would certainly be an interesting question. We have included further references that point to this hypothesis as well and added the following paragraph.

Limitations, outlook and conclusion, page 17, line 385:

“Of particular interest would be the investigation of introspective tasks such as autobiographical memory as these typically induce a positive BOLD response in the PCC, while the coupling with the CMRGlu response is unknown. Such paradigms would also allow to assess whether the presently observed network interactions are symmetrical, i.e., if task positive networks show decreased activation when the DMN exhibits a positive response. This hypothesis seems reasonable in the light of recent work reporting bidirectional information exchange between default mode and other networks (Das et al., 2022; Menon et al., 2023).”

Reviewer #3 (Recommendations for the authors):1) When the authors in the introduction directly talk about "Studies using simultaneous fPET/fMRI" (line 94), I believe fPET and its peculiarities with respect to traditional bolus injection PET, which is the base of most quantitative PET literature, require to be briefly defined since it is not a given that the audience would know this technique.

We thank the reviewer for the thoughtful comment, which will make the work more accessible. In the revised manuscript we have now described the method of fPET in more detail.

Introduction, page 5, line 99:

“In this context, functional PET (fPET) imaging represents a promising approach to investigate the dynamics of brain metabolism. fPET refers to the assessment of stimulation-induced changes in physiological processes such as glucose metabolism (Hahn et al., 2016; Villien et al., 2014) and neurotransmitter synthesis (Hahn et al., 2021) in a single scan. The temporal resolution of this approach of 6-30 s (Rischka et al., 2018) is considerably higher than that of a conventional bolus administration. This is achieved through the constant infusion of the radioligand, thereby providing free radioligand throughout the scan that is available to bind according to the actual task demands. Here, the term “functional” is used in analogy to fMRI, where paradigms are often presented in repeated blocks of stimulation, which can subsequently be assessed by the general linear model.”2

2) Could the authors elaborate on the negative activations shared by BOLD and [18F]FDG in Figures 2 and S1? Some seem to fall for instance in the temporal pole, i.e., in areas that are known to be prone to susceptibility artifacts in fMRI. Are activations in those areas considered to be reliable?

Thank you for highlighting this interesting point. These areas are indeed prone to susceptibility artifacts, however, this is only the case for fMRI acquisitions due to nearby air cavities. Considering that this issue is not present in fPET imaging, we believe that functional PET offers some major opportunities when investigating activation patterns in these regions. This is now included in the manuscript.

Results, page 9, line 190:

“Overlapping negative responses for both imaging modalities occurred in DMN3 (“core”) and DMN4 (“ventral”) for Tetris, but only in DMN4 for working memory (Figure 2f). Simple visual stimulation (eyes open vs. eyes closed) and right finger tapping elicited increased CMRGlu in VIN and SMN (somato-motor), respectively, and a negative response mostly in DMN3 (Figure 2). Notably, some of the regions with negative responses are particularly prone to susceptibility artifacts in fMRI. Since this issue is not present in fPET, these deactivations do not seem to be solely driven by artifacts.”

3) When the anti-correlation between BOLD of task-positive and task-negative networks is brought up, Fox et al. 2005 (https://doi.org/10.1073/pnas.0504136102) might not be the most appropriate reference, as the anticorrelations in that work were largely driven by global signal regression, as later shown by e.g., Murphy et al. 2009 (https://doi.org/10.1016/j.neuroimage.2008.09.036).

We thank the reviewer for the suggestion and agree that another reference is more suitable here. Murphy et al. 2009 found the anticorrelation to be driven by global signal regression and as a consequence, a controversial discussion on this topic emerged. In 2017, Murphy and Fox reviewed this disputed topic together. Overall, we got the impression that Buckner brought this discussion in few words well to a common point:

“…the observed between-network differences are robust. Regardless of whether the anticorrelations are interpreted mechanistically, it is still true that the network linked to external attention is less correlated with the default network than with any other network in the brain“ (Buckner et al., 2019).”

To address this important discussion in our paper, we have added the following sentence to the manuscript, and included more up-to-date references.

Introduction, page 4, line 81:

“Because of the low (or anti-) correlation in BOLD signals between task-positive and default mode networks at resting-state, it has long been assumed that an antagonism between the DMN and other large-scale networks represents a general characteristic of brain functioning (regardless if computed with or without global signal regression (Murphy et al., 2017; Buckner et al., 2019))”

4) The application of a low-pass filter applied to the PET time-activity curves (line 477), despite being mentioned in the authors' previous publications, should be briefly justified.

The rationale for using a low pass filter is to reduce artifacts in the [^18^F]FDG signal that is sampled at a relatively high temporal resolution of 30 s to 1 min frames. More specifically, due to the long task blocks of 6 min (or even longer as in previous work) we assume that changes faster than this do not reflect task-related effects. These unspecific signal changes are therefore reduced by the low-pass filter, which acts as temporal smoothing. This is now included in the manuscript.

Methods, cerebral metabolic rate of glucose metabolism, page 23, line 557:

“Non-gray matter voxels were masked out and a low pass filter was applied, which induces temporal smoothing (cutoff frequency = 3 min). The rationale for this filter is to reduce noise in the high temporal resolution [^18^F]FDG signal. Since task blocks lasted 6 min (or even longer for DS3), we assume that changes faster than this do not reflect task-related effects.”

5) When the Patlak plot is employed (line 486), the chosen t-star (t*) should be reported.6) A reference for the choice of the lumped constant = 0.89 should be provided, as there are controversies over its precise value.

We thank the reviewer for these important details. After visual inspection of representative TACs the value of t* was fixed in the Patlak analysis, which is now included in the text. We also agree with the reviewer that the exact value of the lumped constant is a matter of debate. Still, we would like to emphasize that constant values only induce a scaling of CMRGlu values (see e.g., discussion in Wienhard (2002)). Since this scaling is identical for all subjects it will not affect group statistics. A reference for the lumped constant is now provided in the manuscript.

Methods, cerebral metabolic rate of glucose metabolism, page 24, line 570:

“Quantification was carried out with the Patlak plot (t* fixed to 15 min) and the influx constant K_i_ was converted to CMRGlu as CMRGlu = K_i_ * Glu_plasma_ / LC * 100 with LC being the lumped constant = 0.89 (Graham et al. 2002, Wienhard 2002).”